# Association of Cerebral Venous Thrombosis with mRNA COVID-19 Vaccines: A Disproportionality Analysis of the World Health Organization Pharmacovigilance Database

**DOI:** 10.3390/vaccines10050799

**Published:** 2022-05-18

**Authors:** Jin Park, Moo-Seok Park, Hyung Jun Kim, Tae-Jin Song

**Affiliations:** Department of Neurology, Seoul Hospital, Ewha Womans University College of Medicine, 260 Gonghang-daero, Gangseo-gu, Seoul 07804, Korea; parkjin@ewha.ac.kr (J.P.); strokesolved@ewha.ac.kr (M.-S.P.); khhhj7@ewha.ac.kr (H.J.K.)

**Keywords:** adverse drug reaction, COVID-19, COVID-19 vaccines, venous thrombosis, vaccines

## Abstract

Cerebral venous thrombosis (CVT), a rare thrombotic event that can cause serious neurologic deficits, has been reported after some ChAdOx1 nCoV-19 vaccinations against coronavirus disease 2019 (COVID-19). However, there are few reports of associations between COVID-19 mRNA vaccination and CVT. We retrospectively analyzed CVT occurrence, time of onset after vaccination, outcomes (recovered/not recovered), and death after COVID-19 vaccination from adverse drug reactions (ADR) reports in VigiBase. A disproportionality analysis was performed regarding COVID-19 mRNA vaccines (BNT162b2 and mRNA-1273) and the ChAdOx1 nCoV-19 vaccine. We identified 756 (0.07%) CVT cases (620 (0.05%) after BNT162b2 and 136 (0.01%) after mRNA-1273) of 1,154,023 mRNA vaccine-related ADRs. Significant positive safety signals were noted for COVID-19 mRNA vaccines (95% lower end of information component = 1.56; reporting odds ratio with 95% confidence interval (CI) = 3.27). The median days to CVT onset differed significantly between the BNT162b2 and ChAdOx1 nCoV-19 vaccines (12 (interquartile range, 3–22) and 11 (interquartile range, 7–16), respectively; *p* = 0.02). Fewer CVT patients died after receiving mRNA vaccines than after receiving the ChAdOx1 nCoV-19 vaccine (odds ratio, 0.32; 95% CI, 0.22–0.45; *p* < 0.001). We noted a potential safety signal for CVT occurrence after COVID-19 mRNA vaccination. Therefore, awareness about the risk of CVT, even after COVID-19 mRNA vaccination, is necessary.

## 1. Introduction

Coronavirus disease 2019 (COVID-19) is spreading rapidly [1,2,3,4,5]. Herd immunity is important for the prevention and suppression of the spread of COVID-19, and vaccination is an essential requirement for herd immunity [6]. Two mRNA-based vaccines, BNT162b2 (Pfizer–BioNTech) and mRNA-1273 (Moderna), and a recombinant adenoviral vector vaccine, ChAdOx1 nCoV-19 (AstraZeneca), are currently being administered globally for COVID-19 vaccination [7,8]. Although these vaccines are effective against COVID-19 infection due to their ability to neutralize antibody formations, several side effects related to COVID-19 vaccination have been reported [9,10,11,12]. Notably, COVID-19 vaccination can cause thrombosis and thrombocytopenia through a mechanism related to the production of pathologic antibodies to platelet factor 4 (PF4), leading to systemic venous thrombosis [12,13].

Cerebral venous thrombosis (CVT) is defined as the presence of a blood clot in the cerebral veins or dural venous sinuses. CVT is accompanied by headaches, stroke-related symptoms, and seizures [14]. Hematologic disorders, inflammatory diseases, pregnancy, malignancy, hormonal abnormality, and meningitis are the main associative or risk factors for the development of CVT [14]. The risk factors of CVT after COVID-19 vaccination were similar, with the exception that allergic reactions to the vaccine components and platelet abnormalities were additional factors [15]. Since the initiation of COVID-19 vaccination, cases of CVT associated with COVID-19 vaccines have been reported, with some being fatal [16,17,18,19,20]. Because CVT is a rare disease, case reports or case series of CVT after COVID-19 vaccination are very rare. Moreover, because these reports mainly focus on unusual or interesting cases, it is necessary to investigate the effect of COVID-19 vaccination on real-world CVT occurrence. In addition, although case reports and series have been reported on the association between the ChAdOx1 nCoV-19 vaccine and CVT [16,17,18,19,20], there have been few studies on the associations between mRNA COVID-19 vaccines and CVT.

We hypothesized that COVID-19 vaccines, particularly mRNA-based COVID-19 vaccines, are related to an increased risk of CVT in real-world settings. Therefore, we aimed to perform a disproportionality analysis to investigate the potential safety signals of mRNA-based COVID-19 and ChAdOx1 nCoV-19 vaccines and the clinical characteristics of CVT following these vaccinations using the World Health Organization’s (WHO) global pharmacovigilance database of individual case safety reports: VigiBase.

## 2. Materials and Methods

### 2.1. Study Design and Data Sources

In our study, a disproportionality analysis of adverse drug reactions (ADRs) relative to mRNA-based COVID-19 vaccines and the ChAdOx1 nCoV-19 vaccine was performed using individual case safety reports in VigiBase, WHO’s global deduplicated database, from more than 130 countries [21]. VigiBase is managed by the Uppsala Monitoring Centre (UMC) and has collected information about medication-related ADRs from national pharmacovigilance centers in all countries since 1967. Information about ADRs mainly comes from physicians and various other sources (pharmacists, pharmaceutical companies, or government bodies, such as ministries of food and drug safety). Currently, VigiBase officially provides ADR data for all manufacturers of COVID-19 vaccines but not for each type of COVID-19 vaccine. Furthermore, VigiBase makes no recommendations for comparisons of drugs and states that there are no options for drug–drug or vaccine–vaccine comparisons. Therefore, we analyzed individual case safety reports to identify potential safety signals for CVT after exposure to mRNA-based COVID-19 vaccines and the ChAdOx1 nCoV-19 vaccine compared with the entire database as the control group. Studies using this anonymized, prospectively updated electronic database were approved by the institutional review board of the Ewha Womans University Seoul Hospital (EUMC-2021-08-021). Informed consent was waived because all data were fully anonymized.

### 2.2. Procedures

In our observational case–control study, we extracted all ADR cases of CVT related to mRNA-based COVID-19 vaccines and the ChAdOx1 nCoV-19 vaccine reported in VigiBase using the following preferred terms for CVT in the Medical Dictionary for Drug Regulatory Activities from 20 October 2021 [22]: “transverse sinus thrombosis”, “superior sagittal sinus thrombosis”, “cerebral venous sinus thrombosis”, and “cerebral venous thrombosis”. Information was obtained about age, sex, type of vaccine, time to CVT onset, reporting continents, seriousness, and final outcomes. Seriousness was defined as resulting in significant disability/incapacity, requiring hospitalization, life-threatening, and death. To confirm the differences in the occurrence of CVT by vaccine, the daily numbers of CVT cases were compared over four weeks among the three COVID-19 vaccines.

### 2.3. Disproportionality Analysis

VigiBase uses disproportionality analysis, a case/noncase analysis, to compare ADRs from the entire database with ADRs after COVID-19 vaccinations to detect potential safety signals for vaccine-related ADRs. Disproportionality analysis compares the proportion of ADRs reported for a particular drug to the proportion of ADRs reported in the entire database. If the proportion of ADRs related to COVID-19 vaccinations is greater than those associated with subjects who have not received COVID-19 vaccinations, this suggests potential safety signals of ADRs for COVID-19 vaccination.

Disproportionality is evaluated by calculating the information component (IC) or reporting odds ratio (ROR) using the entire database as a comparator. Detailed methods have been described for the calculation of IC [23,24]. IC calculation was performed using a Bayesian confidence propagation neural network developed and validated by the UMC [21]. Thus, ADR signals for a specific drug can be detected by comparing the possibility of differences in the associated expected and observed drug ADRs from the entire database. IC_025_ is the 95% lower end of the IC. A positive IC_025_ value (> 0) is the threshold for statistical signal detection as defined by the UMC [25]. For sensitivity analysis, we also investigated RORs, which were frequently utilized as potential safety signals before the concept of IC was established [26]. The lower end of the 95% confidence interval (CI) ≥ 1 of RORs with the entire database as the control was defined as the ADR signal detection threshold [27]. Subgroup analyses were performed for the mRNA-based COVID-19 vaccines and the ChAdOx1 nCoV-19 vaccine. As recommended by VigiBase, BNT162b2 and mRNA-1273 were analyzed together as mRNA-based COVID-19 vaccines to reduce selection bias and due to the small sample size.

### 2.4. Statistical Analysis

The characteristics of the subjects were summarized by descriptive statistics; proportions were used for categorical variables and medians with interquartile ranges for non-normally distributed continuous variables, such as time to onset. Categorical variables were compared using the chi-square test or Fisher’s exact test. Differences in time to onset among vaccines were analyzed using the Kruskal–Wallis test with a Bonferroni’s post hoc analysis for intergroup comparison. Two-sided *p* ≤ 0.05 were considered statistically significant. Statistical analyses were performed using R software, version 3.3.3 (R Foundation for Statistical Computing, Vienna, Austria), and SAS 9.4 (SAS Inc., Cary, NC, USA).

## 3. Results

On 30 September 2021, 1513 ADR cases (0.09%) of CVT out of 1,730,636 reports were observed for the mRNA-based COVID-19 vaccines (BNT162b2 and mRNA-1273) and the ChAdOx1 nCoV-19 vaccine. Of these, ADRs of CVT were reported as 756 (0.07%) out of 1,154,023 cases for the mRNA-based COVID-19 vaccines (620 (0.05%) for BNT162b2 and 136 (0.01%) for mRNA-1273) and 757 (0.13%) out of 577,124 cases for the ChAdOx1 nCoV-19 vaccine.

The characteristics of the patients, locations, seriousness, time to onset of CVT after vaccination, and the outcomes of all cases and cases grouped by vaccination are described in Table 1. CVTs were commonly reported in patients aged 18–44 and 45–64 years, more frequently in women, and mainly in Europe and America. In dichotomized age groups of 65 years, there was no significant difference in the reports of CVT between men and women (*p* = 0.16). The median time to onset (days) of CVT was significantly different between the BNT162b2 and ChAdOx1 nCoV-19 vaccines (median 12 (interquartile range 3–22) vs. median 11 (interquartile range 7–16), respectively, *p* = 0.02). The differences in the time to onset within 28 days of vaccination, grouped by vaccine, are presented in Figure 1. There were significant differences in the time to onset among all vaccines (*p* = 0.03). However, post hoc analysis revealed significant differences only between the BNT162b2 and ChAdOx1 nCoV-19 vaccines (*p* = 0.02). Appendix A shows the daily numbers of CVT cases and the cumulative frequency for all vaccines for the entire period. The time to onset of CVT was significantly earlier in the age group less than 65 years than that in people older than 65 years (95% CI, −10.82–−2.76; *p* = 0.001), but there was no significant difference between sexes (95% CI, −0.06–6.33; *p* = 0.05) (Appendix A).

More than 90% of the patients were in serious condition, and 33% did not recover or died. The outcome of death after CVT was significantly higher in patients who received the ChAdOx1 nCoV-19 vaccine than in those who received the mRNA-based COVID-19 vaccines (odds ratio (OR) = 0.32; 95% CI, 0.22–0.45; *p* < 0.001). In pairwise comparisons of the different types of vaccines, vaccination with ChAdOx1 nCoV-19 more often led to death after CVT than vaccination with BNT162b2 (OR = 0.35; 95% CI, 0.25–0.50; *p* < 0.001) or mRNA-1273 (OR = 0.18; 95% CI, 0.07–0.44; *p* < 0.001) (Appendix A).

A significant signal of disproportionality of CVT was noted for all COVID-19 vaccines (IC_025_ = 2.01; ROR_025_ = 5.14) and separately for the mRNA-based COVID-19 vaccines (IC_025_ = 1.56; ROR_025_ = 3.27) and the ChAdOx1 nCoV-19 vaccine (IC_025_ = 2.56; ROR_025_ = 6.70) with respect to IC_025_ and ROR (Figure 2).

## 4. Discussion

The key findings of our study of CVT cases from VigiBase reported by 130 countries are that the potential safety signal for the development of CVT was noted in mRNA-based COVID-19 vaccines as well as the ChAdOx1 nCoV-19 vaccine compared with the entire dataset.

There are few reports on CVT after mRNA-based COVID-19 vaccination [17,28]. These studies suggested that CVT occurrences related to mRNA-based COVID-19 vaccines may be due to endothelial dysfunction caused by spike glycoprotein interactions with endothelial cells resulting in immunothrombosis. If the spike glycoprotein of mRNA-based COVID-19 vaccines binds to the angiotensin-converting enzyme 2 receptor, several inflammatory and thrombogenic molecules, such as leukocyte chemotactic factors, cell adhesion molecules (vascular cell adhesion molecule 1 and intercellular adhesion molecule 1), and procoagulant cytokines, can be activated. This mechanism may cause endothelial dysfunction, particularly in brain endothelial cells [29], which could contribute to a significant disruption of brain endothelial barrier integrity, ultimately promoting thrombus formation. Moreover, a previous study suggested that the spike glycoprotein may induce platelet aggregation and activation and eventually result in thrombus formation [30]. Although the period of time in which the spike glycoprotein persists has not been clearly established, several studies have suggested that it may last for weeks. Thus, spike glycoprotein-related platelet activation triggered by mRNA-based COVID-19 vaccines could explain the trend of CVT occurrences after mRNA-based COVID-19 vaccinations [30,31]. Furthermore, in line with these previous case reports, our results showed that CVT occurred mainly within a few weeks of mRNA-based COVID-19 vaccinations.

In agreement with the well-known relationship between the ChAdOx1 nCoV-19 vaccine and CVT [32,33], our results showed a potential safety signal of the ChAdOx1 nCoV-19 vaccine for CVT. Additionally, in one recent study, about 90% of CVT cases following COVID-19 vaccination occurred after the administration of the ChAdOx1 nCoV-19 vaccine [15]. The ChAdOx1 nCoV-19 vaccine can cause systemic thromboembolism due to thrombosis with thrombocytopenia syndrome [34]. The Food and Drug Administration has found a causal link between the adenovirus vector COVID-19 vaccine and thrombosis with thrombocytopenia syndrome and has provided updates on rare clotting or thrombotic events following adenovirus vector COVID-19 vaccination, primarily in young women [35]. Clinical courses and laboratory test results suggest that the pathogenesis of thrombosis with thrombocytopenia syndrome is similar to that of autoimmune heparin-induced thrombocytopenia. Autoimmune heparin-induced thrombocytopenia is caused by the formation of antibodies to PF4, a component of platelet alpha granules released during platelet activation. Unlike classical heparin-induced thrombocytopenia, endogenous polyanions, such as chondroitin sulfate or polyphosphate, may trigger PF4 antibody formation in autoimmune heparin-induced thrombocytopenia [19,35]. Other previous studies and reports by the European Medicines Agency have demonstrated that an adverse immune reaction called immunosenescence may occur in young people, leading to disseminated intravascular coagulation-like blood changes after ChAdOx1 nCoV-19 vaccination [36,37,38,39]. Our research supports the evidence for the risk of CVT after ChAdOx1 nCoV-19 vaccination with real-world data.

Interestingly, our results showed that there was a difference in the onset of CVT after exposure to the mRNA-based COVID-19 vaccines and ChAdOx1 nCoV-19 vaccine. The median values of 13 and 11 days for the time to onset of CVT for the mRNA-based COVID-19 vaccines and the ChAdOx1 nCoV-19 vaccine, respectively, were similar. However, the mRNA-based COVID-19 vaccines had the highest number of CVT cases in the first week after vaccination, after which the incidence decreased gradually. In contrast, the ChAdOx1 nCoV-19 vaccine showed the highest incidence of CVT in the second week after vaccination and a sharp decrease thereafter. These timelines suggest that the thrombosis mechanisms of these vaccines differ. The mRNA-based COVID-19 vaccines, which target the spike glycoprotein of SARS-CoV-2, directly induce intracellular production of the spike protein. This spike protein plays a key role in the initiation of the immune response, which may last for up to a few weeks. CVT occurred from 1 to 9 days after vaccination [17,28], and the neutralizing antibody titer after vaccination was maintained for 35 to 119 days [40,41]. This evidence supports a wide temporal distribution of CVT occurrence related to mRNA-based COVID-19 vaccination in our study. In contrast, the formation of PF4 antibodies and PF4–polyanion complexes in vaccine-induced immune thrombotic thrombocytopenia, the most well-known thrombotic complication after ChAdOx1 nCoV-19 vaccination, takes time [11]. A type-II heparin-induced thrombocytopenia that develops via a similar mechanism occurs 5 to 14 days after exposure due to the time required for the formation of antibodies [42]. In another study of the ChAdOx1 nCoV-19 vaccine, CVT was reported 5 to 30 days (median 14 days) after vaccination [43]. Regardless, it is possible that other mechanisms of thrombosis have not yet been elucidated.

In this study, the time until CVT occurrence after vaccination was significantly different between the BNT162b2 and ChAdOx1 nCOV-19 vaccines but not between the mRNA-1273 and ChAdOx1 nCOV-19 vaccines. Various factors, such as vaccine components or immune responses, might cause this difference between mRNA vaccines. However, our study could not provide acceptable evidence related to these differences. Recently, several studies that compared the BNT162b2 and mRNA-1273 vaccines reported the possibility of differences in clinical responses or outcomes in addition to SARS-CoV-2 antibody responses [44,45]. The rates of breakthrough infections and 60-day hospitalizations were significantly lower in those vaccinated with mRNA-1273 compared with BNT162b2; this result suggests that these vaccines may act via different mechanisms [45]. Further studies on the specific mechanism of CVT occurrence after mRNA vaccination and prospective studies on clinical outcomes are needed.

In our study, the number of deaths in CVT patients was lower after mRNA vaccination than after vaccination with ChAdOx1 nCoV-19. A previous study showed that significant risk factors for mortality due to thrombosis with thrombocytopenia syndrome after ChAdOx1 nCoV-19 vaccination were intracerebral hemorrhage and CVT [46]. However, there have been few studies on CVT occurrence after mRNA vaccination and CVT-related mortality. Furthermore, it is difficult to present accurate evidence because VigiBase does not provide information on various parameters related to mortality, such as laboratory data, brain-imaging findings, or the occurrence of systemic thromboembolism.

Our study has limitations. First, if the national drug-monitoring center of a country does not report ADRs, these cases will not be present in VigiBase. However, VigiBase includes rare ADRs and generalized ADR information from more than 130 countries. Second, VigiBase does not provide any validation of laboratory findings, radiologic information, or accuracy of diagnosis. Information on whether CVT occurred after the first or the second vaccine dose was also not included. Third, vaccine-induced immune thrombotic thrombocytopenia received major public attention after April 2021 [13], and this may have affected the increased reports of CVT cases after COVID-19 vaccination. Lastly, as mentioned above, it is difficult to directly compare outcome parameters, including death, between the different types of COVID-19 vaccines in VigiBase.

## 5. Conclusions

Our study demonstrated a potential safety signal for occurrence of CVT for COVID-19 mRNA vaccination. It is necessary to be aware of the risk of CVT occurrence, even after COVID-19 mRNA vaccination.

## Figures and Tables

**Figure 1 vaccines-10-00799-f001:**
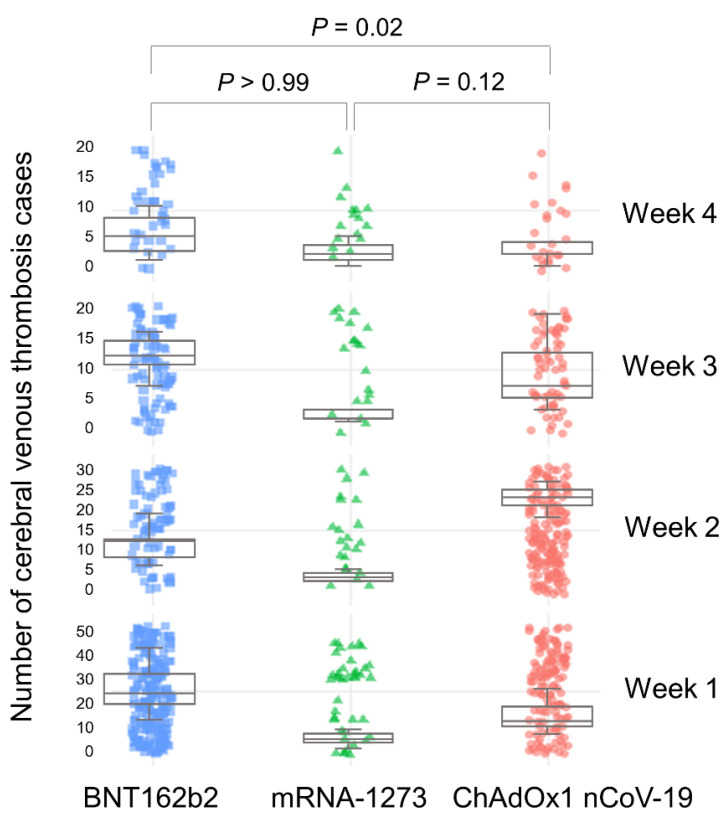
Differences between times of occurrence of cerebral venous thrombosis within 28 days after vaccine administration for each COVID-19 vaccine type. The box plots indicate median numbers of cerebral venous thrombosis cases and interquartile ranges per week depending on each vaccine. There are significant differences in time to cerebral venous sinus thrombosis onset from date of vaccination between the different vaccines within 28 days. In pairwise comparisons between vaccine groups, only significant differences between BNT162b2 and ChAdOx1 nCoV-19 vaccines remained (*p* = 0.02).

**Figure 2 vaccines-10-00799-f002:**
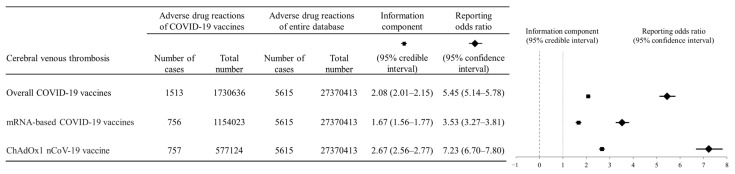
Disproportionality analysis between mRNA-based vaccines and the ChAdOx1 nCoV-19 vaccine to compare cerebral venous thrombosis occurrence in vaccinated individuals with the entire VigiBase database. The information component (IC) and reporting odds ratio (ROR) were calculated for the disproportionality analysis. In this forest plot, overall COVID-19 vaccines showed significantly positive associations with cerebral venous thrombosis by IC_025_ (2.01) and ROR_025_ (5.14).

**Table 1 vaccines-10-00799-t001:** Demographics and characteristics of reported cases of cerebral venous thrombosis according to type of COVID-19 vaccine.

Characteristics	Total (*N* = 1513)	BNT162b2 (*N* = 620)	mRNA-1273 (*N* = 136)	ChAdOx1 nCoV-19 (*N* = 757)
Age, years				
≤11	2 (0.1)	0 (0.0)	0 (0.0)	2 (0.3)
12–17	12 (0.8)	12 (1.9)	0 (0.0)	0 (0.0)
18–44	479 (31.7)	161 (26.0)	54 (39.7)	264 (34.9)
45–64	482 (31.9)	129 (20.8)	40 (29.4)	313 (41.3)
65–74	185 (12.2)	60 (9.7)	26 (19.1)	99 (13.1)
≥75	124 (8.2)	71 (11.5)	11 (8.1)	42 (5.5)
Unknown	229 (15.1)	187 (30.2)	5 (3.7)	37 (4.9)
Sex				
Male	574 (37.9)	229 (36.9)	65 (47.8)	280 (37.0)
Female	928 (61.3)	386 (62.3)	71 (52.2)	471 (62.2)
Unknown	11 (0.7)	5 (0.8)	0 (0.0)	6 (0.8)
Continentals				
Africa	1 (0.1)	1 (0.2)	0 (0.0)	0 (0.0)
Americas	447 (29.5)	325 (52.4)	98 (72.1)	24 (3.2)
Asia	7 (0.5)	3 (0.5)	0 (0.0)	4 (0.5)
Europe	1006 (66.5)	286 (46.1)	38 (27.9)	682 (90.1)
Oceania	52 (3.4)	5 (0.8)	0 (0.0)	47 (6.2)
Seriousness				
Yes	1439 (95.1)	568 (91.6)	129 (94.9)	742 (98.0)
No	74 (4.9)	52 (8.4)	7 (5.1)	15 (2.0)
Time to onset (day)	12 (5.0–21.0)	12 (3.0–22.0)	15 (4.5–27.5)	11 (7.0–16.0)
Outcome				
Recovered	267 (17.6)	76 (12.3)	21 (15.4)	170 (22.5)
Recovered with sequelae	316 (20.9)	175 (28.2)	53 (39)	88 (11.6)
Recovering	369 (24.4)	182 (29.4)	38 (27.9)	149 (19.7)
Not recovered	319 (21.1)	101 (16.3)	13 (9.6)	205 (27.1)
Death	184 (12.2)	44 (7.1)	5 (3.7)	135 (17.8)
Unknown	58 (3.8)	42 (6.8)	6 (4.4)	10 (1.3)

Data are presented as numbers (%) or medians (interquartile range). Seriousness: resulting in significant disability/incapacity, requiring hospitalization, life-threatening, and death. Time to onset (days): calculated time to onset of cerebral venous thrombosis. Based on vaccination date and adverse drug reaction start date, expressed as medians and interquartile ranges. Unknown: cases for which information was unavailable from VigiBase.

## Data Availability

Publicly available datasets were analyzed in this study. These data can be found here: https://vigiaccess.org/ (accessed on 30 September 2021).

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
