# Peer review of "Association of Cerebral Venous Thrombosis with mRNA COVID-19 Vaccines: A Disproportionality Analysis of the World Health Organization Pharmacovigilance Database"

_vaccines, 2022, doi:10.3390/vaccines10050799_

Round 1
Reviewer 1 Report
In the current study, the authors attempted to analyze the association between CVT and COVID-19 vaccination using the WHO pharmacovigilance database. In agreement with the well-known relationship between CVT and the ChAdOx1 nCoV-19 vaccine, they confirmed the adenoviral-vectored vaccine was related with an increased risk of CVT occurrence in the real-world data. Notably, they have found that the potential safety signal for development of CVT was noted in mRNA-based COVID-19 vaccines as well as the ChAdOx1 nCoV-19 vaccine. This is a well-written paper containing important information. However, I raised two points which need to clarify. These are given below.
Specific comments:
- What could be the reason why the mortality rate of CVT patients who received the mRNA vaccine is much lower than the mortality rate of CVT patients who received ChAdOx1 nCoV-19? The authors should discuss it in the discussion section.
- In Fig. 1, there were significant differences between BNT162b2 and ChAdOx1 nCOV-19 vaccines (P = 0.02). In contrast, no difference was observed between mRNA-1273 and ChAdOx1 nCOV-19 vaccines (p = 0.12). BNT162b2 is basically the same type of vaccine as mRNA-1273. I wonder what seems to have caused this difference.
Author Response
Answer to comments
Manuscript ID: vaccines-1681648
Title: Association of cerebral venous thrombosis with mRNA COVID-19 vaccines: a disproportionality analysis of the World Health Organization pharmacovigilance database
Reviewer 1.
In the current study, the authors attempted to analyze the association between CVT and COVID-19 vaccination using the WHO pharmacovigilance database. In agreement with the well-known relationship between CVT and the ChAdOx1 nCoV-19 vaccine, they confirmed the adenoviral-vectored vaccine was related with an increased risk of CVT occurrence in the real-world data. Notably, they have found that the potential safety signal for development of CVT was noted in mRNA-based COVID-19 vaccines as well as the ChAdOx1 nCoV-19 vaccine. This is a well-written paper containing important information. However, I raised two points which need to clarify. These are given below.
Specific comments:
- What could be the reason why the mortality rate of CVT patients who received the mRNA vaccine is much lower than the mortality rate of CVT patients who received ChAdOx1 nCoV-19? The authors should discuss it in the discussion section.
⇒ Thank you for your insightful suggestion. The mortality of CVT patients after COVID-19 vaccination may be influenced by various predisposing factors. Unfortunately, we only compared the reports of death among the vaccines due to the limitation of the study based VigiBase. In our study, the peak time of CVT occurrence after exposure to each vaccine was different between the mRNA vaccine (first week) and ChAdOx1 nCoV-19 vaccine (second week). And this might be the clue that suggests the different mechanisms of CVT between mRNA vaccines and ChAdOx1 nCoV-19 vaccine. We hope that the need for further studies on the specific mechanism of CVT after mRNA vaccines and prospective studies on the mortality and clinical outcomes draw more attention. We clarify the limitation of comparison such as the death of CVT cases after COVID-19 vaccination in this study using VigiBase in the Discussion as follows:
(Last two paragraphs of the Discussion section) “In our study, the number of death cases in CVT patients who received the mRNA vaccines was much lower than in the same instances of the CVT patients who received ChAdOx1 nCoV-19. A previous study showed high mortality (35.9%) of thrombosis with thrombocytopenia syndrome after ChAdOx1 nCoV-19 vaccination, and significant risk factors of mortality were intracerebral hemorrhage and CVT.[1] However, there are few studies on the CVT occurrence after mRNA vaccination and CVT related mortality. Furthermore, it is difficult to present exact evidence as VigiBase does not provide information on various parameters related to mortality, such as laboratory data, brain imaging findings, or the occurrence of systemic thromboembolism.
Our study has limitations. First, if the national drug monitoring center of a country does not report ADRs, these cases will not be present in VigiBase. However, VigiBase includes rare ADRs and generalized ADR information from more than 130 countries. Second, VigiBase does not provide any validation of laboratory findings, radiologic information, or accuracy of diagnosis. The information on the first or the second dose was also not included. Third, vaccine-induced immune thrombotic thrombocytopenia received major public attention after April 2021[2] and this may affect the increased reporting of the cases of CVT after COVID-19 vaccination. Lastly, as mentioned above, it is difficult to directly compare the outcome parameters including death according to the different types of COVID-19 vaccines in VigiBase.”
- Hwang J, Park SH, Lee SW, Lee SB, Lee MH, Jeong GH, et al. Predictors of mortality in thrombotic thrombocytopenia after adenoviral COVID-19 vaccination: the FAPIC score. Eur Heart J. 2021;42(39):4053-63. Epub 2021/09/22. doi: 10.1093/eurheartj/ehab592.
- Pascual-Iglesias A, Canton J, Ortega-Prieto AM, Jimenez-Guardeño JM, Regla-Nava JA. An Overview of Vaccines against SARS-CoV-2 in the COVID-19 Pandemic Era. Pathogens. 2021;10(8). Epub 2021/08/29. doi: 10.3390/pathogens10081030.
- In Fig. 1, there were significant differences between BNT162b2 and ChAdOx1 nCOV-19 vaccines (P = 0.02). In contrast, no difference was observed between mRNA-1273 and ChAdOx1 nCOV-19 vaccines (p = 0.12). BNT162b2 is basically the same type of vaccine as mRNA-1273. I wonder what seems to have caused this difference.
⇒ We appreciated the comment that you mentioned. The finding that there were fewer mRNA-1273 cases compared to other vaccine groups may have influenced the results of the statistical analysis. In the Supplementary figure that depicts the distribution of the number of CVT occurrences after vaccination over the entire period, the pattern up to day 28 was similar between BNT162b2 and mRNA-1273 but different for ChAdOx1 nCOV-19 though the P value was not significant (P = 0.12). In addition, when comparing BNT162b2 and mRNA-1273, the P value was >0.99 which reflects almost no difference between these vaccines.
In a recent study, the possibility of different clinical responses or outcomes in the mRNA vaccines was reported. [Reference No. 39] The rate of breakthrough infection (HR, 0.85; 95% CI, 0.80-0.89) and 60-day hospitalizations (HR, 0.80; 95% CI, 0.70-0.91) were significantly lower in mRNA-1273 and this result suggests that these vaccines may have different detailed mechanisms. We additionally describe in the Discussion as follows:
(The fifth paragraph of the Discussion) “In this study, the time of CVT occurrence after vaccination was significantly different between BNT162b2 and ChAdOx1 nCOV-19, but not between mRNA-1273 and ChAdOx1 nCOV-19. Various factors such as components of each vaccine or immune responses might cause this difference between mRNA vaccines. However, we cannot raise acceptable evidence on this difference within the scope of our study. Recently, several studies that compared BNT162b2 and mRNA-1273 reported the possibility of differences in different clinical responses or outcomes as well as SARS-Cov-2 antibody response.[1, 2] The rate of breakthrough infection and 60-day hospitalizations were significantly lower in mRNA-1273 compared to BNT162b2, and this result suggests that these vaccines may have different detailed mechanisms.[2] Further studies on the specific mechanism of CVT after mRNA vaccines and prospective studies on the mortality and clinical outcomes need more attention.”
- Steensels D, Pierlet N, Penders J, Mesotten D, Heylen L. Comparison of SARS-CoV-2 Antibody Response Following Vaccination With BNT162b2 and mRNA-1273. JAMA. 2021;326(15):1533-5. Epub 2021/08/31. doi: 10.1001/jama.2021.15125.
- Lindsey Wang, Pamela B. Davis, David C. Kaelber, Nora D. Volkow, Rong Xu. Comparison of mRNA-1273 and BNT162b2 Vaccines on Breakthrough SARS-CoV-2 Infections, Hospitalizations, and Death During the Delta-Predominant Period. JAMA. 2022;327(7):678-680. doi:10.1001/jama.2022.0210
We sincerely appreciate your invaluable comments.

Reviewer 2 Report
In this timely study, the authors performed a disproportional analysis to investigate the occurrence of cerebral venous thrombosis in subjects vaccinated with COVID-19 mRNA vaccine. They found that the potential for the development of CVT was noted in the mRNA based COVID-19 vaccines as well as the ChAdOx1 nCoV-19 vaccine compared to the entire ADR dataset. Thus this knowledge can help in early diagnosis and treatments can have a positive outcome for CVT patients.
Even though these are interesting data, the authors did not mention about the other risk factors that might have contributed to the development of CVT. Also, the study findings that the cases of CVT are mainly in Europe and America (Is it North and South combined?) could be due to the higher rate of vaccination.
Overall, this is a well-written manuscript and I recommend it be accepted for publication after minor revisions.
- Describe the risk factors that that might have contributed to the development of CVT.
- Line 45- cases of CVD associated with COVID-19 vaccine
- In the Table 1, specify Unknown
- In Figure 1 (Line 159), it’s not clear which week the statistically significant differences remain between BNT162b2 and ChAdOx1 nCoV-19 vaccines (P = 0.02)
- Line 160- modify sentence- More than 90% of patients were in serious conditions…
Author Response
Answer to comments
Manuscript ID: vaccines-1681648
Title: Association of cerebral venous thrombosis with mRNA COVID-19 vaccines: a disproportionality analysis of the World Health Organization pharmacovigilance database
Reviewer 2.
In this timely study, the authors performed a disproportional analysis to investigate the occurrence of cerebral venous thrombosis in subjects vaccinated with COVID-19 mRNA vaccine. They found that the potential for the development of CVT was noted in the mRNA based COVID-19 vaccines as well as the ChAdOx1 nCoV-19 vaccine compared to the entire ADR dataset. Thus this knowledge can help in early diagnosis and treatments can have a positive outcome for CVT patients.
Even though these are interesting data, the authors did not mention about the other risk factors that might have contributed to the development of CVT. Also, the study findings that the cases of CVT are mainly in Europe and America (Is it North and South combined?) could be due to the higher rate of vaccination.
Overall, this is a well-written manuscript and I recommend it be accepted for publication after minor revisions.
- Describe the risk factors that might have contributed to the development of CVT.
⇒ Thank you for your comment. We described additionally the risk factors for the development of CVT after COVID-19 vaccination in the Introduction as follows:
“Hematologic disorders, inflammatory diseases, pregnancy, malignancy, hormonal abnormality, and meningitis are the main association or risk factors for the development of CVT.[1] The risk factors of CVT after COVID-19 vaccination were similar to the general population except for a history of allergy to any of the vaccine components or platelet abnormality.[2]”
[1] Stam, J. Thrombosis of the cerebral veins and sinuses. N Engl J Med 2005, 352, 1791-1798, doi:10.1056/NEJMra042354.
[2] de Gregorio, C.; Colarusso, L.; Calcaterra, G.; Bassareo, P.P.; Ieni, A.; Mazzeo, A.T.; Ferrazzo, G.; Noto, A.; Koniari, I.; Mehta, J.L.; et al. Cerebral Venous Sinus Thrombosis following COVID-19 Vaccination: Analysis of 552 Worldwide Cases. Vaccines 2022, 10, 232. https://doi.org/10.3390/vaccines10020232
- Line 45- cases of CVD associated with COVID-19 vaccine
⇒ Thank you for your careful comment. We add the word based on your recommendation.
“…, cases of CVT associated with COVID-19 vaccine …”
- In the Table 1, specify Unknown
⇒ Thank you for the thorough comment. We add the explanation of “Unknown” at the end of the table as below:
“Unknown: the cases that information was unavailable from the VigiBase Database.”
- In Figure 1 (Line 159), it’s not clear which week the statistically significant differences remain between BNT162b2 and ChAdOx1 nCoV-19 vaccines (P = 0.02)
⇒ Thank you for your insightful comment. For intuitive understanding, Figure 1 showed the distribution of time of occurrence of CVT within 28 days after vaccine administration per week and by vaccine type. The P-value was compared between the vaccines for the entire 28 days from exposure to the vaccines. Each median of CVT cases with bar plots was presented by week, however, pairwise comparison was not performed by a week due to the possibility of the statistic bias from the decreased number of each group, especially the small number of mRNA-1273. We add the time for comparison in the figure captions for clarification as below:
“There are significant differences in time to cerebral venous sinus thrombosis from date of vaccination among the different vaccines within 28 days.”
- Line 160- modify sentence- More than 90% of patients were in serious conditions…
⇒ Thank you for your recommendation. Based on your recommendation, the text has been changed as follows:
“More than 90% of the patients were in serious conditions”
We sincerely appreciate your invaluable comments.
